# Polarized X-ray scattering measures molecular orientation in polymer-grafted nanoparticles

Subhrangsu Mukherjee [1], Jason K. Streit[2,3], Eliot Gann[1], Kumar Saurabh[4], Daniel F. Sunday [1], Adarsh Krishnamurthy [4], Baskar Ganapathysubramanian[4], Lee J. Richter[1], Richard A. Vaia [2] & Dean M. DeLongchamp [1]✉

Polymer chains are attached to nanoparticle surfaces for many purposes, including altering solubility, influencing aggregation, dispersion, and even tailoring immune responses in drug delivery. The most unique structural motif of polymer-grafted nanoparticles (PGNs) is the high-density region in the corona where polymer chains are stretched under significant confinement, but orientation of these chains has never been measured because conventional nanoscale-resolved measurements lack sensitivity to polymer orientation in amorphous regions. Here, we directly measure local chain orientation in polystyrene grafted gold nanoparticles using polarized resonant soft X-ray scattering (P-RSoXS). Using a computational scattering pattern simulation approach, we measure the thickness of the anisotropic region of the corona and extent of chain orientation within it. These results demonstrate the power of P-RSoXS to discover and quantify orientational aspects of structure in amorphous soft materials and provide a framework for applying this emerging technique to more complex, chemically heterogeneous systems in the future.

[1] Material Measurement Laboratory, National Institute of Standards and Technology, Gaithersburg, MD, USA. [2] Materials and Manufacturing Directorate, Air Force Research Laboratory, Wright Patterson Air Force Base, OH, USA. [3] UES, Inc., Dayton, OH, USA. [4] Department of Mechanical Engineering, Iowa State University, Ames, IA, USA. ✉email: dean.delongchamp@nist.gov

Despite decades of advancement in structural characterization methods, the molecular orientation in nanoscale regions of organic materials remains surprisingly inscrutable. Although regions with crystalline order are readily measured using diffraction-based methods, amorphous regions are not. Conventional scattering probes of polymer structure, such as hard X-rays, neutrons, and electrons, are not directly sensitive to local orientation correlations of organic groups. Many phenomena that make polymers attractive for engineering applications occur in these difficult-to-measure amorphous regions, such as plastic and elastic deformation and the transport of charge, ions, and molecules.

Questions about the structure within polymer nanocomposites, and polymer-grafted nanoparticles (PGNs) in particular, present a classic case-in-point[1–4]. PGNs are a vibrant frontier of polymer physics, where questions of confinement, entanglement, and phase behavior are explored[5–7]. A key feature of modern PGN research is a focus on the impact of graft density and how it affects the conformation of chains arranged radially about the particle[6,8,9]. At high graft densities, it is expected that a concentrated polymer brush (CPB) region will form close to the particle surface. Chains are expected to be highly stretched or extended within this brush region. Despite the importance of local chain extension, in descriptions of PGNs both experimentally[10,11] and computationally[4], there have been no direct measurements of the conformation and orientation of the brush itself. Almost everything that is known is inferred from indirect measurements. The most common approach to characterizing the chain behavior is to measure the hydrodynamic particle radius, examine its scaling with chain length, and calculate the dimensions of various regions based on arguments of excluded volume and statistical segment length. Even heroic efforts[10,12,13] using small-angle neutron scattering (SANS) can only measure the compositional distribution of the brush, and the conformation is inferred from that using sophisticated models. The orientation of stretched chains in the amorphous, nanoscale CPB region has not been measured due to a lack of orientation-sensitive measurement techniques.

Here, we make the direct measurements of chain orientation in a PGN CPB. We use polarized resonant soft X-ray scattering (P-RSoXS), an emerging X-ray scattering technique. Unlike conventional small-angle X-ray scattering (SAXS) with hard X-rays, and SANS, P-RSoXS imparts a unique sensitivity to molecular orientation through the interaction of near-edge X-ray absorption fine structure (NEXAFS) transition dipoles with incident X-ray polarization[14–21]. Unlike the simple scalar treatments of SAXS and SANS, quantitative solutions of P-RSoXS patterns are rare due to the complexity of describing the orientation fields. In this manuscript, we advance the interpretation framework for P-RSoXS by further developing computational scattering pattern simulation using real-space structure models. We achieve a quantitatively solved P-RSoXS result describing both the spatial extent of grafted chain orientation in a CPB and its magnitude. The success of this approach in measuring previously unmeasurable aspects of the polymer structure demonstrates that P-RSoXS has matured into a powerful general tool for nanoscale characterization challenges in amorphous soft matter. We anticipate that it will soon be possible to develop structure-property relationships between chain orientation, graft density, and excluded volume for a variety of chain chemistries and explore local chemistry-factors affecting polymer chain conformation such as dihedral potentials[3]. The ability to measure molecular orientations may enable a better understanding of the underlying physics which drives nanoparticle self-assembly[22,23] and provide another step toward programmable nanoparticle morphologies. With recent advances made to P-RSoXS sample environments to include liquids[24], it should soon be possible to probe the influence of solvents. More broadly, P-RSoXS promises great potential for quantitative measurement of orientational aspects of structure in amorphous soft materials that were previously inaccessible[15]. Potential targets abound where the chemical specificity of P-RSoXS to select specific functional groups[25] and measure their orientation independent of other functional groups could prove especially powerful, for example, in electrical transduction[26], solar energy conversion[27], renewable nanocomposites[27], smart materials[28], and structural biology[29].

## Results and discussion

Polystyrene (PS) grafted gold nanoparticles (AuNP, radius ≈ 10 nm) were prepared and cast into monolayer films using established methods[7]. Two samples having different number averaged molecular mass for the grafted polymer chains (≈27 and ≈53 kDa) and different graft densities estimated at $\sigma \approx$ 1.2 chains/nm$^2$ and ≈0.6 chains/nm$^2$, respectively, were prepared. The two samples shall be henceforth referred to as AuPS27 and AuPS53, respectively. A representative AFM image of the AuPS27 film is shown in Fig. 1a, which reveals a disordered hexagonal array of nanoparticles with an average interparticle separation of ≈(35–40) nm. An AFM image of the AuPS53 sample that has a larger interparticle separation of ≈(45–50) nm is shown in Figure S1a. Films were cast onto silicon│silicon nitride substrates that were later etched[30] to provide windows for P-RSoXS.

**P-RSoXS experimental results**. The central physical principle of P-RSoXS is that resonant absorption of soft X-rays causes changes in the real and imaginary parts of the index of refraction near the resonant energy, and scattering patterns are therefore highly dependent on energy and polarization. At non-resonant energies, the scattering pattern is isotropic (Figure S2), whereas at 285.2 eV, an energy near the PS (1s→π*) resonance, the pattern exhibits anisotropy (Fig. 1b) that is induced by the polarization of the incident X-rays, with more intense scattering perpendicular to the electric field vector. Sector-averaged 1-D profiles from sectors parallel (∥) and perpendicular (⊥) to the incident electric field (Fig. 1c) show a strong first-order structure factor peak due to a paracrystalline 2D hexagonal close-packed lattice at $q \approx$ 0.15 nm$^{-1}$ and weaker higher-order peaks, consistent with hard X-ray SAXS collected on similar films[7]. The occurrence of anisotropy across these peaks at energies near the PS (1s→π*) resonance indicates local correlations in the molecular orientation of the aromatic rings within the polymer corona. In P-RSoXS analysis it is common to analyze trends in anisotropy at an incident photon energy $E$ via a parameter, $A$, derived from differences in the scattered intensity perpendicular and parallel to the electric field vector, i.e.,

$$A(q) = \left. \frac{I(q)_\parallel - I(q)_\perp}{I(q)_\parallel + I(q)_\perp} \right|_E ;$$ this value can be integrated over measured $q$

and shown as a function of energy, i.e., $A(E) = \left. \frac{I_\parallel - I_\perp}{I_\parallel + I_\perp} \right|_E$, where $I_{\parallel or \perp} = \int_{q_{min}}^{q_{max}} I(q) dq$, as in Fig. 1d[14,16,17]. We confirmed that $A(q)$ was identical for two orthogonal directions of the electric field of the incident X-rays (Figure S3), demonstrating the absence of any nonisotropic/non-uniaxial assembly in the films and the average of the two $A(q)$ profiles corresponding to the two electric field polarizations at each X-ray energy was used for all the analysis in this study[8]. The sign of $A(E)$ switches at ≈285 eV, with extrema occurring at 284.7 and 285.2 eV. Therefore, the anisotropy is entirely due to the interaction of X-ray polarization and local chain orientation in the CPB. The above observations were found to be qualitatively similar for the AuPS53 sample (Figure S1). However, its strong first-order structure factor peak is at a lower $q$-value (≈0.13 nm$^{-1}$), corresponding to a larger interparticle

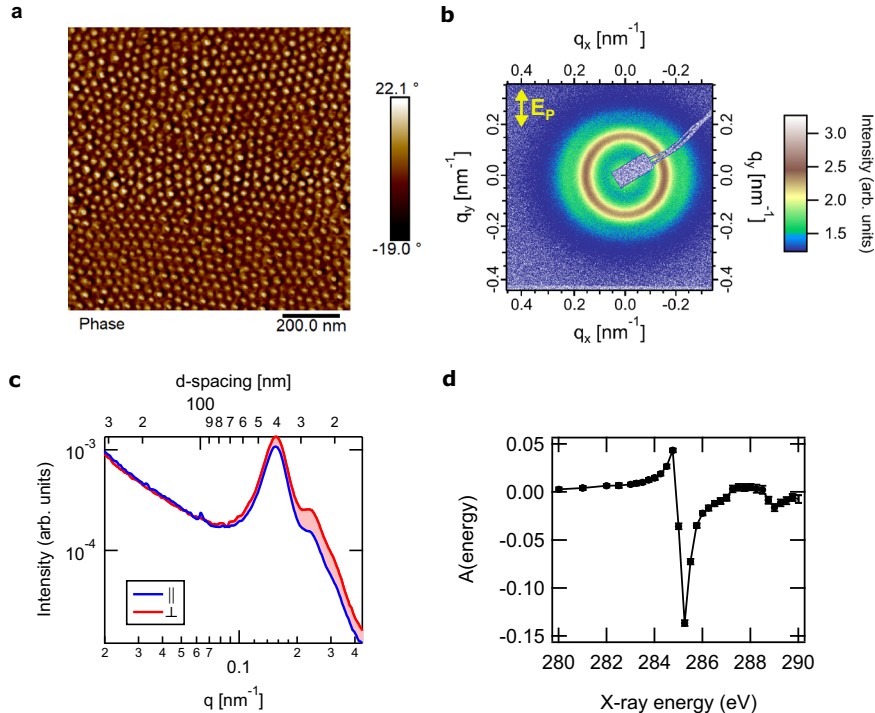

**Fig. 1 Microscopy of, and scattering from, AuPS27 nanoparticle ultrathin film. a** AFM phase image showing hexagonally close-packed nanoparticle array. **b** Anisotropic scattering pattern obtained using linearly polarized soft X-ray photons with the polarization vector of the electric field (**E_P**) along the vertical direction at a resonant energy for the aromatic rings in PS, 285.2 eV. **c** Radial scattering profiles parallel (∥) and perpendicular (⊥) to incident X-ray polarization extracted from scattering pattern shown in (**b**). Area of shaded region between the curves is proportional to the magnitude of scattering anisotropy. Red (blue) shading implies negative (positive) anisotropy. **d** Anisotropy parameter (see text for definition) calculated by integrating the scattering profiles over the experimental $q$-range (0.02–0.4 nm$^{-1}$).

separation consistent with the AFM data. While the energy and $q$-dependence of the two samples are similar, the magnitude of anisotropy ($A(E)$ and $A(q)$) was considerably lower for the AuPS53 sample. Next, we develop a computational scattering pattern simulation and fitting approach to extract quantitative molecular orientation information from this rich experimental data set.

**Index of refraction tensor**. A critical requirement for P-RSoXS pattern simulation of oriented soft material is an energy-dependent 3D tensor that connects the structure and degree of orientation of the molecule to the real and imaginary parts of the tensor the index of refraction, $n = 1 - \delta + i\beta$ (or scattering length density, with simple transforms). Energy dependence itself is straightforward, as PS is readily measured by NEXAFS spectroscopy[14,31,32], providing the imaginary part ($\beta$), and a Kramers–Kronig consistent calculation provides the real part ($\delta$)[33]. However, breaking the real and imaginary parts into a 3D tensor requires consideration of the molecular geometry.

Our P-RSoXS measurement exhibits the greatest anisotropy at energies near ≈285 eV (Fig. 1d), for which the only NEXAFS transition is the PS phenyl (1s→π*), and thus it is the phenyl rings that are oriented. Although crystalline syndiotactic PS has phenyl rings that are comprehensively oriented relative to the backbone[34,35], our PS is atactic and amorphous. In Fig. 2a, we show an image of atactic PS where the backbone is fully extended with an all-anti conformation. For each phenyl ring, the 1s→π* transition dipole moment is perpendicular to the ring plane[31], as shown in Fig. 2b, and each sp³ bond attaching the phenyl to the backbone is 90° from an all-anti backbone long axis. We then assume that the phenyl groups may adopt any fixed rotation about this sp³ attachment point due to amorphousness. An

ensemble of 1s→π* transition dipoles will therefore describe a plane perpendicular to the sp³ attachment bond, as shown in Fig. 2c. Finally, we account for random stereochemistry and amorphousness by assuming that all rotations about the backbone axis are equally possible. Integration of this arrangement yields the uniaxial representation for the absorption (imaginary part) shown in Fig. 2e, which can be intuitively understood from Fig. 2d as all of the planes contribute to the direction parallel to the backbone axis, but they do not all contribute to the other two axis directions. This uniaxial tensor has an extraordinary axis parallel to the backbone with an increase in 1s→π* absorbance relative to isotropic PS; the ordinary axes orthogonal to the backbone have a decrease in 1s→π* relative to isotropic PS. The corresponding $\delta$ and $\beta$ are shown in Figure S4.

**Morphology model**. Our computational approach to P-RSoXS pattern simulation uses a real-space morphology model composed of voxels[14]; in this case a 512 × 512 × 32 volume with each voxel cube having 2.5 nm sides. Each voxel is composed of a material that has an amorphous and anisotropic fraction. If the material is anisotropic, it has a vector describing the orientation of the extraordinary axis of our uniaxial index tensor. Because the structure factor is such a prominent feature of the scattering pattern, it was important to correctly locate the AuNP centers. Rather than simulating the paracrystalline PGN arrangement[7], we applied an image analysis algorithm[36] to the measured AFM images, which located every particle detectable by the eye. We then placed voxels having a gold index of refraction at each particle center location to construct the AuNPs. In a core-shell type morphology, radially oriented anisotropic coronas were then generated around each AuNP, such that the extraordinary axis of the index was radial with respect to the particle center. The

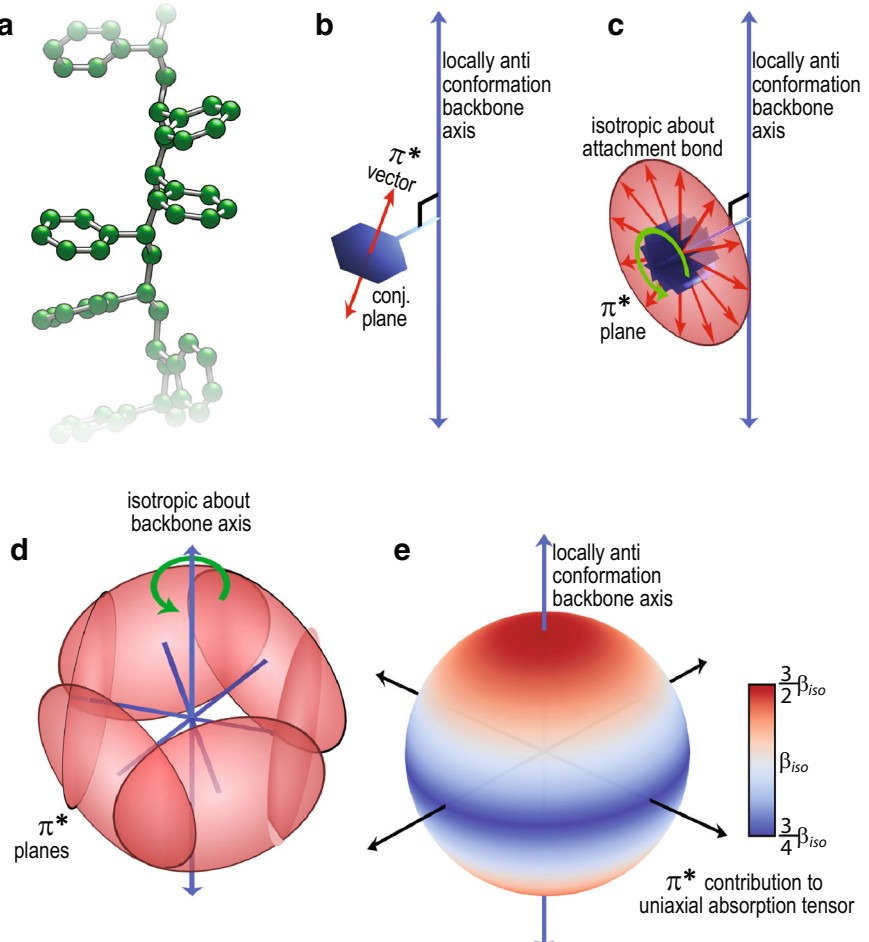

**Fig. 2 Development of the index of refraction tensor originating from the orientation distribution in the PS aromatic ring. a** An idealized PS chain with all-anti backbone conformation. **b** The $1s \rightarrow \pi^*$ transition dipole moment is normal to the phenyl ring plane and the $sp^3$ attachment bond is 90° to the backbone. **c** An ensemble of random rotations about the $sp^3$ attachment bond causes the $1s \rightarrow \pi^*$ transition dipole moment to describe a plane. **d** An ensemble of rotations about the backbone axis accounts for amorphousness and atacticity. **e** An integrated uniaxial absorption tensor ($\beta$) for the $1s \rightarrow \pi^*$ resonance under these assumptions.

thickness of the anisotropic corona ($r_{aniso}$) was varied, and the remaining voxels were isotropic PS. The orientation in the corona was defined by two variables: $S_0$ and $d$. $S_0$ is an orientational order parameter applied to the layer of PS corona immediately adjacent to the particle surface, with a value of 1 for perfect radial orientation and 0 for isotropic. Orientation within the corona decays as $S(r) = S_0 \left( \frac{r_{np}}{r} \right)^d$, where $r$ is the distance from the nanoparticle center and $r_{np}$ is the nanoparticle radius; $d = 2$ for orientation decaying proportionally to cross-sectional area. Example morphologies are shown in Fig. 3. For brevity, we will confine our discussion to the AuPS27 system; for the AuPS53 system, the same approach was used with different inputs.

**Pattern simulation**. To compare a simulated P-RSoXS pattern to the experiment, we must first select a model anisotropic canopy thickness ($r_{aniso}$). A reasonable starting point is the critical radius given by Ohno et al.[11]:

$$r_c = \frac{r_0 \sqrt{\sigma_0^*}}{\upsilon^*} \qquad (1)$$

where $r_0$ is the AuNP radius, $\sigma_0^*$ is the dimensionless graft density, and $\upsilon^*$ is a rescaled excluded volume parameter, as given by Daoud and Cotton[37]. The measured graft density in AuPS27 is

1.2 chains/nm², consistent between thermogravimetric analysis and ultraviolet-visible spectroscopy. This is very close to the theoretical limits (see SI), which suggests that it also includes the presence of free chains in equilibrium with the PGNs. This graft density, and graft densities far smaller, such as that of AuPS53, deliver the same conclusion: the CPB is thicker than the interparticle separation. Thus, the entire corona should be considered CPB. If we assume that $r_{aniso} \approx r_c$ then $r_{aniso}$ must encompass the whole corona. With an average interparticle separation of ≈40 nm and $r_0 \approx 10$ nm, $r_{aniso} \approx 12.5$ nm reasonably fills the interparticle volume, as shown in the third column of Fig. 3. We simulated P-RSoXS patterns of the AuPS27 morphology, using the index tensor developed in Fig. 2, and $S_0 = 1$, $d = 2$. Simulations were performed using a graphics processing unit (GPU) accelerated forward simulator based on a previously published[14] sequential approach.

The simulated 2D scattering pattern compares excellently to the experimental one, being anisotropic and more intense perpendicular to the electric field vector at 285.2 eV (Figure S6). Sector-averaged 1-D profiles extracted from the 2-D patterns from sectors parallel ($\parallel$) and perpendicular ($\perp$) to the incident electric field also agree well with experiment (Fig. 1c). The peaks originating from the structure factor are faithfully recreated in position and relative intensity, confirming that the Fig. 1a AFM from which the morphological models were developed sufficiently

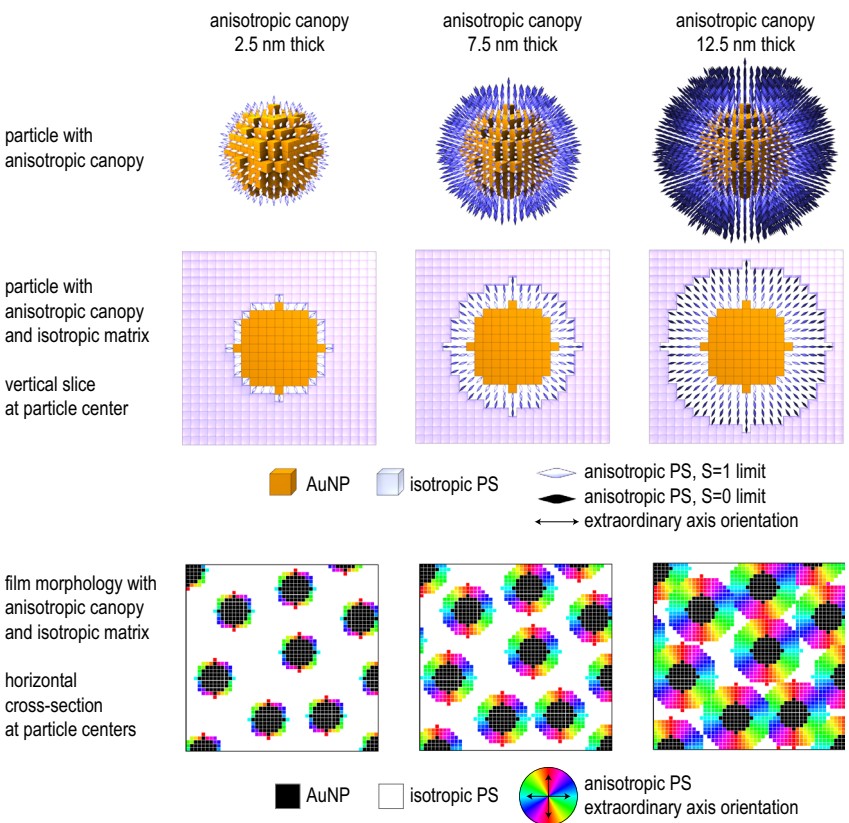

**Fig. 3 Proposed morphology models for different anisotropic canopy thicknesses ($r_{aniso}$) with orientational order parameter, $S_0 = 1$.** First and second rows show isolated particles; third row shows magnified whole-film morphology. Orientation decays as $d = 2$. Voxels in all images are 2.5 nm cubes. AuNP gold nanoparticle, PS polystyrene.

represents the sample within the beam spot ($\approx 200\ \mu m$), and that the interparticle separations measured by AFM are a reasonable representation of the particle spacings in the volume measured by P-RSoXS. However, the $A(q)$ extracted from simulated patterns had a higher intensity (higher anisotropy) and a peak position at a lower $q$-value compared to the experiment. Since tangential chains are sometimes reported[38], we further checked a model with chains tangential to the particle and found that the simulated pattern is more intense parallel to the electric field vector (Figure S7), which is the opposite of the experiment. Thus, a net tangential chain orientation is inconsistent with the experimental results, although it is certainly possible that some tangential chains are present. Whether a minority fraction of voxels could have tangential chain orientation with a majority fraction of voxels having radial chain orientation was left unexplored; our models assume a homogeneous orientation that is radially symmetric about the particle center to avoid overfitting. Grafted chain orientational heterogeneity on a $\approx 2$ nm length scale is likely best explored using realistic molecular physics models and consistency with P-RSoXS measurements could be used to validate such models.

**Fitting simulations to experiment.** The basic structure of the model is therefore sound, but details of the anisotropic corona require adjustment. Our fast P-RSoXS simulations on GPUs enable a high-throughput multi-resolution parametric sweep of the 3-parameter system. The goal of the fit was to minimize the sum of squared errors (SSE) between the measured and simulated $A(q)$ by varying $r_{aniso}$, $S_0$, and $d$. A coarse-grained pass of the parameter space explores the full SSE landscape and identifies promising low SSE regions. A second pass then performs a fine resolution sweep of these promising regions to identify the best fit

to experiment. Because the $A(q)$ representation appears sensitive to anisotropy details of the corona, we calculated a mean squared error between experiment and simulation for $A(q)$ at 284.7 and 285.2 eV. Results of this approach for the AuPS27 sample are shown as 2-D heatmaps in Fig. 4. A clear minimum is observed in the parameter space which is further reinforced by uncertainty quantification analysis (see SI for details). The optimum for the $r_{aniso}$ and $S_0$ parameter pair was narrow, whereas it was broader for plots including $d$. Changes to $r_{aniso}$ affected the location of the peak in $A(q)$, and changes to $S_0$ affected the magnitude of $A(q)$ (Figures S10 and S11), whereas changes to $d$ affected both (Figure S12), leading to some correlations between $r_{aniso}$ and $d$, and $S_0$ and $d$, but clear fits were still found. The morphological model with $r_{aniso} = 7.4$ nm, $S_0 = 0.5$, and $d = 0.4$ is the best fit to the experimental data for the AuPS27 sample. The decay $d$ is substantially lower than 2, suggesting that orientation persists away from the particle surface more than might be expected based on a cross-sectional area argument ($\sim \frac{1}{r^2}$). When additional parameters (besides the above three—$r_{aniso}$, $S_0$, and $d$) were included in the model to achieve more realistic morphologies such as core size variation (from SEM, Figure S13) and surface roughness (from AFM) we observed a marginal decrease in SSE (1.3 to 1.2) without noticeable change in the fits. The one-sided nature of the error distribution (Figure S16) obtained from the fits to the experimental data using the 5-parameter model (Figure S17) shows that the optimal values obtained from the 3-parameter model are still valid for the 5-parameter model.

To determine whether the P-RSoXS pattern anisotropy arises from the orientation in the corona of individual particles, or from the orientation correlations between the corona of neighboring particles, simulations were performed on morphologies consisting of different number of PGNs. Starting from only two particles spaced

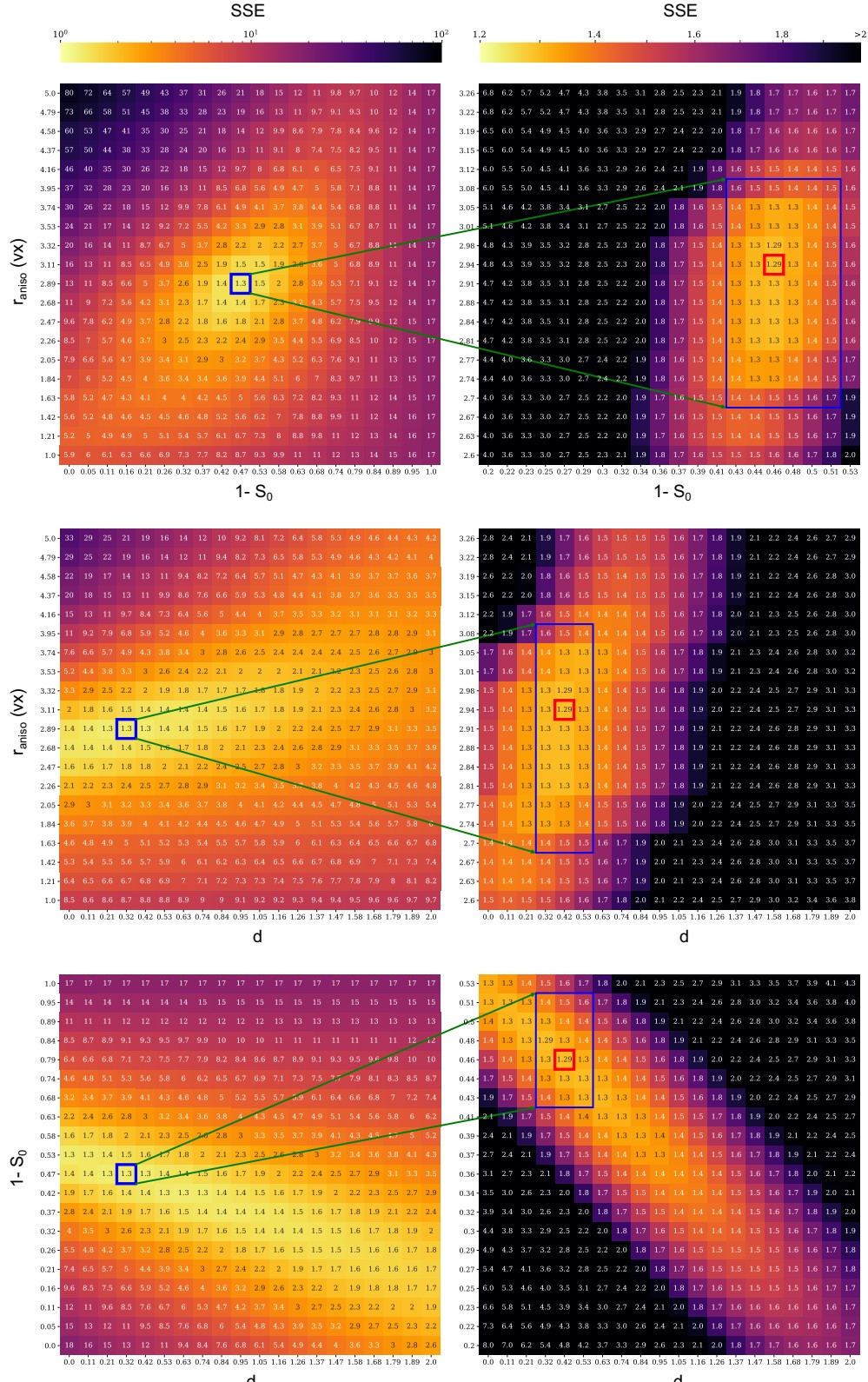

**Fig. 4 2-D heatmaps of total sum of squared errors (SSE) between experiment and simulation for $A(q)$.** The total SSE for $A(q)$ were calculated at the two energies (284.7 and 285.2 eV) across 16,000 models having different parameters ($r_{aniso}$, $S_0$, and $d$) for the anisotropic corona that were used to fit the simulations to the experiment. Number within each pixel shows the residual sum of squares corresponding to each model. $r_{aniso}$ is given in voxels (1 vx = 2.5 nm). While the search was performed in the 3D ($r_{aniso}$, $d$, $S_0$) space, we show 2D slices across the optimal values for ease of visualization. For the definition of $r_{aniso}$, $S_0$, and $d$, see text.

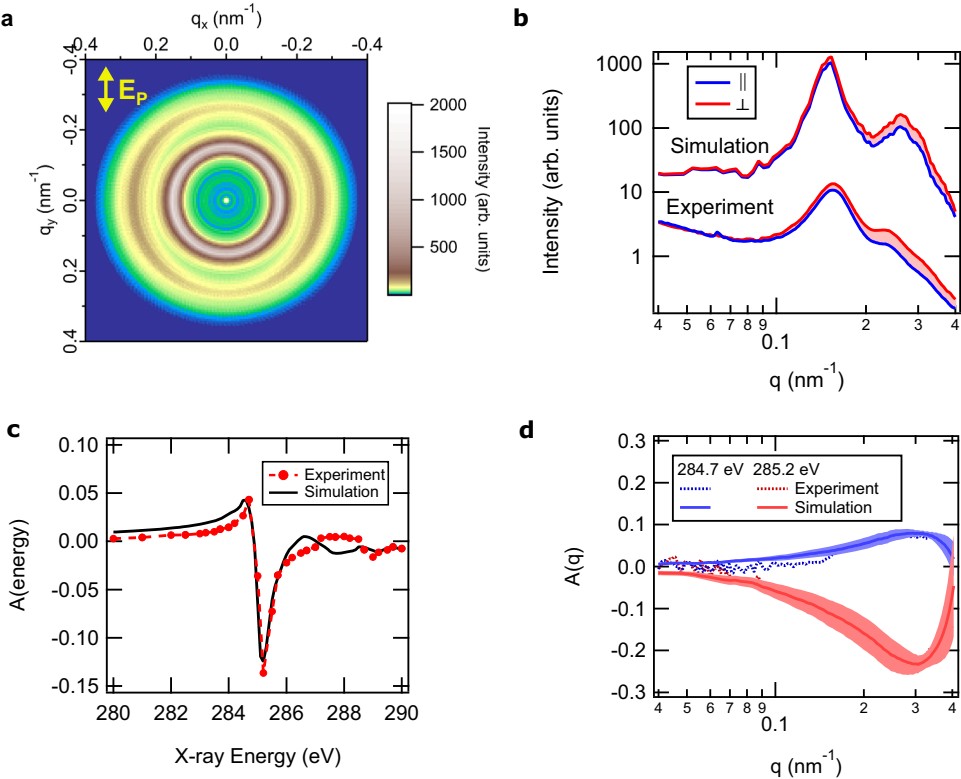

**Fig. 5 Simulation results for $r_{aniso}$ = 7.4 ± 0.2 nm, $S_0$ = 0.5 ± 0.1, $d$ = 0.4 ± 0.5.** **a** Simulated anisotropic scattering pattern at a resonant energy of 285.2 eV with the polarization vector of the electric field ($E_P$) along the vertical direction. **b** Radial scattering profiles parallel (∥) and perpendicular (⊥) to incident X-ray polarization extracted from the simulated scattering pattern shown in (**a**). Area of shaded region between the curves is proportional to the magnitude of scattering anisotropy. Red (blue) shading implies negative (positive) anisotropy. **c** Experimental versus simulated anisotropy ratio ($A$(energy)) calculated by integrating the scattering profiles over the $q$-range (0.02–0.4 nm$^{-1}$). **d** $q$-dependence of anisotropy parameter at 284.7 and 285.2 eV. Shaded region in (**d**) is the error in calculation of $A(q)$ obtained from uncertainty quantification (see SI) of the three parameters ($r_{aniso}$, $S_0$, $d$) in P-RSoXS simulations.

far apart so that the interparticle separation is larger than total particle size, the particle locations were filled up randomly to finally achieve the final simulated morphology for the AuPS27 sample. The three model parameters were kept fixed at $r_{aniso}$ = 6.8 nm, $S_0$ = 0.5, and $d$ = 0 for these simulations. The effect of the number of particles on the scattering features is shown in Figure S18. The structure factor peak at $q \approx 0.16$ nm$^{-1}$ becomes evident when there are ≥50 particles in the morphology. In the dilute limit (2–20 particles) the scattering pattern is still anisotropic, as seen in the $A$($q$) functions shown in Figure S19. Within the experimentally accessible $q$-range (0.02–0.4 nm$^{-1}$) the variation in $A(q)$ with number of particles is negligible. The strongest differences are observed at the $q$-locations corresponding to the form factor minima in the dilute limit. These differences would be due to interparticle correlations and measuring them might be possible in the future with lower-noise detectors. However, under real-world conditions with particle size variation, wavelength spread, and slit smear, such effects could be subtle. These simulation results show that $A(q)$ vs. $q$ has special value as an objective function for fitting P-RSoXS data because in densely packed systems it is largely insensitive to the structure factor. In model morphologies where $I(q)$ vs. $q$ does not completely resemble the experimental data (because, perhaps, a single AFM image is not completely representative of the ensemble measured by scattering), fitting $A(q)$ vs. $q$ will still extract orientation parameters relevant to orientation in the corona of individual particles.

**Best match with the experiment**. P-RSoXS simulation results for the best fit model with $r_{aniso}$ = 7.4 ± 0.2 nm, $S_0$ = 0.5 ± 0.1, and

$d$ = 0.4 ± 0.5 are shown in Fig. 5. Figure 5a, b retain excellent similarity to the experimental structure factor. The excellent agreement between experiment and simulation in Fig. 5c, d confirms that these parameters accurately describe the oriented region within the anisotropic corona. It is consistent with CPB theory that our measured $r_{aniso}$ is far less than $r_c$. The $r_c$ value is based on the linearity of the relationship between chain contour length and hydrodynamic radius for a fixed graft density. CPB theory does not require that all chains within it be fully stretched, in fact it is expected that hydrodynamic radius will be significantly smaller than would be expected for fully stretched chains[11], and the fully stretched condition is anticipated only immediately adjacent to the particle surface[39]. Within the CPB, one would expect a continuous decrease in $S$ away from the particle surface.

**Comparison of graft density and molecular weight**. The same fitting approach was applied to the AuPS53 data, with a best fit of $r_{aniso}$ = 9.3 ± 0.8 nm, $S_0$ = 0.4 ± 0.1, and $d$ = 2 (Figure S20). The results for the two samples are summarized in Fig. 6. Compared to AuPS27, the AuPS53 order parameter is lower, its oriented region thickness is slightly larger, and its orientation decay is now proportional to increasing cross-sectional area. The lower $S_0$ and higher $d$ in AuPS53 indicate a substantially reduced amount of overall chain orientation, which is consistent with its graft density being roughly half of what was measured in AuPS27. The orientation parameters for the two samples can also be understood based on the expected interaction of the grafted and free chains. When the graft density is ~1 chain/nm$^2$ the entropically

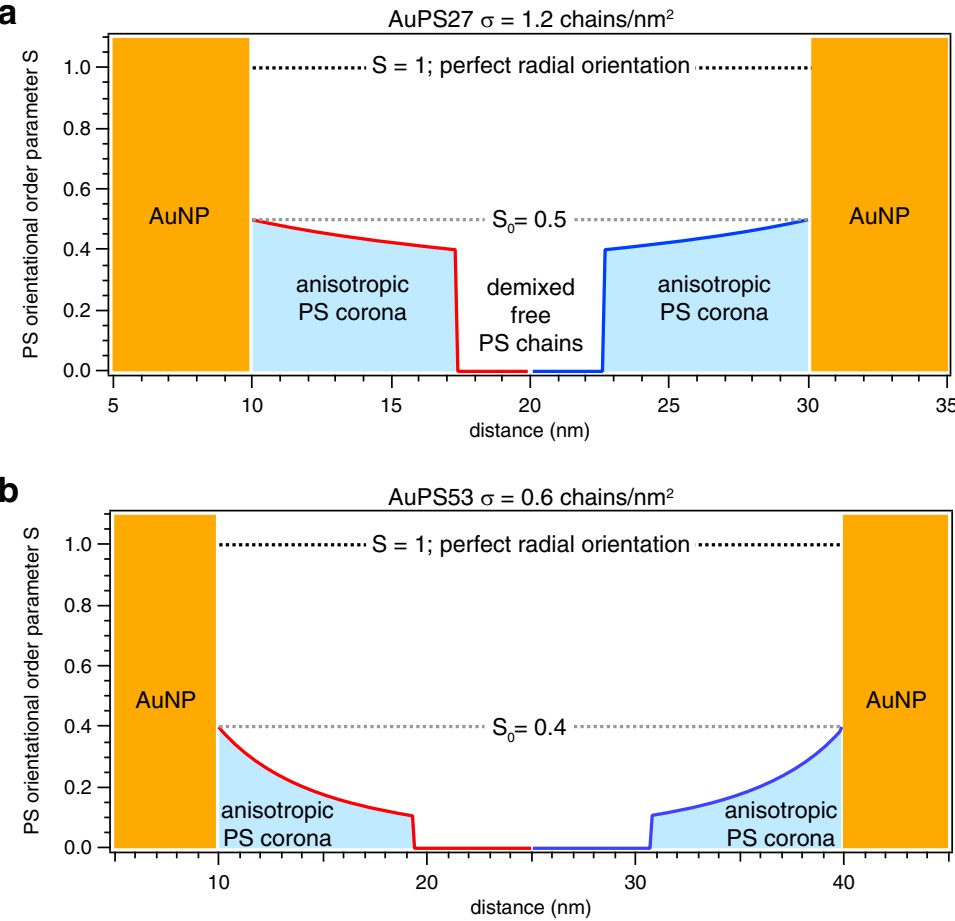

**Fig. 6 Schematic representation of corona structure obtained from fits to A(q). a** AuPS27, and **b** AuPS53 samples with different graft densities ($\sigma$). The gold nanoparticles (AuNP) are shown as yellow columns and the blue-shaded regions between the nanoparticles denote the anisotropic polystyrene (PS) corona.

driven demixing transition referred to as autophobic dewetting will occur when the free chain length is equivalent or greater than the grafted chain length[40,41]. Therefore, in the AuPS27 sample, the free chains will be expelled from the corona due to autophobic dewetting, which results in the tight region of highly oriented chains around the particle surface that abruptly drops off into a field of isotropic PS. For AuPS53 the graft density is lower, and the ratio of the free/grafted chain length places the sample within the wetting regime. This will result in significant intermixing of the free chains with the grafted corona causing the grafted chains to extend away from the particle surface. This result is consistent with the larger oriented region measured for AuPS53. The difference in the entropic interactions with the free chains may also account for the different decay rates of the oriented layers. Our results here are also in excellent agreement with recent studies[42,43] based on coarse-grained molecular dynamics simulations that predict a substantial interpenetration layer within which chain section conformations are unperturbed. The larger thickness of the oriented region and decay order of two for the AuPS53 are consistent with a lack of orientational perturbation in this corona overlap region. Future studies may be able to directly incorporate such predictions of molecular simulations with R-SoXS model generation algorithms to directly compare experiment, simulation, and theory.

We must note that our results are based on some key assumptions. In the dielectric constant model developed in Fig. 2, we assume that the phenyl groups, being amorphous, may adopt

any rotation about the long axis of the polymer backbone. If there is any preference for the phenyl groups relative to the long axis, the absolute value of our measured orientational order parameter ($S_0$) would change. However, the spatial extent of the orientation, and the trend in relative $S_0$ with molecular mass would be unaffected. Secondly, we assume that the AFM is representative of the bulk structure; the appearance of structure factors similar to experiment in the simulated pattern appears to strongly support this assumption. Finally, we assume that the response surface is smooth enough that the likely minimum is identified using our multi-resolution algorithm. A combination of modern imaging techniques[44], optical constants from accurately predicted NEXAFS spectra[45], measurements on cryotomed bulk samples, liquid environments and modern adaptive fitting algorithms[46] may someday overcome these limitations and such studies are currently underway.

In summary, using polarized resonant soft X-ray scattering (P-RSoXS) we have made quantitative and nanoscale-resolved measurements of chain orientation within the polymer corona of PS-grafted Au nanoparticles. Our fitting analysis of P-RSoXS measurements allows us to separate the thickness of the anisotropic region of the corona from the extent of orientation within it. Significant differences are observed between particle systems having different PS graft density. Higher graft density results in higher orientation near the particle surface and a much greater overall extent of orientation. The thiol-based chemistry used to prepare the particles results in a significant fraction of

demixed PS chains in the high graft density system, and the spatial extent of that demixed region can also be measured. P-RSoXS measurements of orientation within the anisotropic corona of a PGN provide a new window into the behavior of polymer chains in confinement and the impacts of that confinement on molecular orientation.

## Methods

**Polymer-grafted nanoparticles (PGN) film preparation**. A flow-coater with a motorized translational stage was used to prepare thin films of polymer-grafted nanoparticles on silicon | silicon nitride (56 nm) substrates. The angle and gap height between the blade and substrate were 5° and 250 μm, respectively. Approximately 10 μL of PGN dispersion in toluene was deposited between blade and surface. The substrate was moved via a translation stage at constant speed of 10 mm/s and air-dried. Concentration of the PGN deposition solution was determined with ultraviolet-visible (UV−vis) spectroscopy (Cary 300 Bio, Varian) using the extinction coefficient ($7.1 \times 10^8$ $M^{-1}.cm^{-1}$) at the gold surface plasmon peak of 530 nm based on the size of the nanoparticles[7,47]. The coated films were placed in a sealed holder with an etch window and the backside of the substrates were anisotropically etched in a KOH etch solution to expose the transparent nitride windows following a procedure[30] described below.

**Patterning of backside nitride and anisotropic etch in KOH**. S1813 photoresist was spin-coated at 314 rad/s (3000 rpm) on 400-μm-thick Si wafers (100 mm diameter, (100) orientation) (NOVA Electronic Materials) coated with ≈56 nm silicon nitride, and annealed at 115 °C for 90 s. The resist was then exposed on a SUSS MA6 mask aligner to define windows and alignment marks on the backside of the wafer. The patterned wafers were then developed in MF-319 developer for 90 s and rinsed with deionized (DI) water. The exposed nitride was then dry etched with CF4 plasma. The remaining resist was stripped by keeping the wafers immersed in MICROPOSIT Remover 1165 overnight at 80 °C and a final rinse with DI water. After the PGN films were coated on the front (unpatterned) side the wafers were placed in a wet etch holder (from AMMT GmbH) with the front side sealed off by O-rings to protect the PGN film and only the patterned backside exposed to 25% (mass/volume) KOH aqueous solution at 70 °C. The etch was stopped after the silicon backing the windows was fully etched. The holder was then removed from the etch solution, and rinsed with DI water. The wafers were then removed from the holder, and dried.

**Polarized resonant soft X-ray scattering (P-RSoXS)**. P-RSoXS measurements were performed in transmission geometry at the Advanced Light Source (ALS) beamline 11.0.1.2[48]. The 2-D scattering patterns were collected using a Peltier cooled (−45 °C) in-vacuum charge-coupled device (CCD) detector (PI-MTE, Princeton Instruments, 2048 × 2048 pixels). Data were collected at selected energies across the C K-edge (270 eV to 290 eV) with two different incident X-ray polarizations (horizontal and vertical with respect to the lab frame). The P-RSoXS data were analyzed using a custom version of the Nika[49] package based on Igor Pro.

**Atomic force microscopy (AFM)**. AFM images were recorded on the same films (on nitride membranes) used for P-RSoXS measurements using a Bruker Dimension Icon atomic force microscope operating in tapping mode.

**Multi-resolution fit and uncertainty quantification**.

1. Coarse-grained parameter search:

Parameter space:

 I. $r_{aniso} = [2.5 \text{ nm}, 12.5 \text{ nm}]$
 II. $S_0 = [0.0, 1.0]$
 III. $d = [0.0, 2.0]$

Number of points spanned for each parameter: 20 points uniformly sampled. Results: The sum of squared errors (SSE) minimum was found at $r_{aniso} = 7.2$ nm, $S_0 = 0.5$, and $d = 0.3$.

2. Fine-grained parameter search:

The parameter range corresponding to ±15% of the SSE minimum from the coarse-grained parameter search was selected for the second stage. The resultant parameter space:

 I. $r_{aniso} = [6.5 \text{ nm}, 8.2 \text{ nm}]$
 II. $S_0 = [0.4, 0.8]$
 III. $d = [0.0, 2.0]$

Number of points spanned for each parameter: 20 points uniformly sampled. The SSE minimum at the second stage was found for $r_{aniso} = 7.4$ nm, $S_0 = 0.5$, and $d = 0.4$.

Uncertainty in model parameters from fits:

The variance-covariance matrix[50] and the uncertainty in the three parameters for the AuPS27 sample were calculated from the heat map data shown in Fig. 4. The matrix (Figure S21) was calculated from samples with SSE below an appropriately chosen threshold. The choice of the SSE threshold was such that if the latter is raised, the increase in the variance/covariance values is much less compared to the increase in number of samples and the mean vector for all the samples equals the best fit values for the three parameters.

**Scanning electron microscopy (SEM)**. To determine average AuNP core size, high-resolution scanning electron microscopy (SEM) images were taken of AuPS27 and AuPS53 monolayer films deposited on Si substrates. SEM images were acquired using a Gemini 500 field emission SEM (Carl Zeiss AG, Oberkochen, Germany) at an acceleration voltage of 5 kV. A representative image of AuPS27 monolayer film is shown in Figure S13a. Custom image analysis software written using Matlab 2018b (Mathworks) was applied to determine the radius of each AuNP in the image. A small number of outliers consisting of nanoparticle dimers were automatically excluded from the analysis. The resulting distribution of AuNP radii is shown in Figure S13b, giving an average radius of (9.9 ± 1.0) nm.

**Thermogravimetric analysis (TGA)**. Thermogravimetric analysis (TGA) was used to calculate graft density, $\Sigma$, where $\Sigma = \left(\frac{w_{PS}}{100 - w_{PS}}\right)\frac{N_A \rho_{NP}}{3\pi r_0 M_n}$ and $w_{PS}$, $N_A$, $\rho_{NP}$, $r_0$, and $M_n$ are the experimental mass loss of PS-grafted NP, Avogadro's number, the density of gold (19.3 g/cm$^3$), AuNP radius, and the number averaged molecular weight of PS, respectively. TGA measurements were conducted under flowing $N_2$ at a heating rate of 10 °C/min using a TA Hi-Res TGA 2950 thermogravimetric analyzer. An example of an experimental TGA curve showing the weight loss of PS at ≈400 °C is shown in Figure S22.

## Data availability

The processed anisotropy data, optical constants, and particle location data for the AuPS27 sample are available in Figshare database at https://doi.org/10.6084/m9.figshare.14851050.v2.

## Code availability

All codes for analyses herein are available upon request.

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

## Acknowledgements
Certain commercial equipment, instruments, materials, or software are identified in this paper in order to specify the experimental procedure adequately. Such identification is not intended to imply recommendation or endorsement by NIST, nor is it intended to imply that the materials or equipment identified are necessarily the best available for the purpose. P-RSoXS data were acquired at the Advanced Light Source, which was supported by the Director, Office of Science, Office of Basic Energy Sciences, of the U.S. Department of Energy under Contract DE-AC02-05CH11231. Beamline support at beamline 11.0.1.2 was provided by A.L.D. Kilcoyne and C. Wang. S.M. gratefully acknowledges B. Robert Ilic (NIST) for his assistance with the preparation of the silicon nitride substrates. B.G. and K.S. acknowledge Department of the Navy, Office of Naval Research grant N00014-19-12453.

## Author contributions
S.M., R.A.V., and D.M.D. conceived the idea and designed the experiments and analysis. S.M. did the P-RSoXS experiments and performed all data analysis. J.K.S and R.A.V. made the PGN samples. E.G., K.S., A.K., and B.G. developed the scattering simulation code. S.M. and D.M.D. wrote the morphology generation code that enabled fitting simulations to experiment. K.S., A.K., and B.G. did the multi-resolution fits. D.F.S. took the AFM data. L.J.R. developed the model for the index of refraction tensor. S.M. and D.M.D. wrote the manuscript. All authors discussed the experiments, edited the manuscript, and gave consent for this publication.

## Competing interests
The authors declare no competing interests.
