## [Peer Review File · Nature Communications]

REVIEWER COMMENTS

Reviewer #1 (Remarks to the Author):

This manuscript reports on developing an approach based on polarized resonant soft X-ray scattering (P-RSoXS) to examine the local alignment of chains grafted to nanoparticle surfaces. The combination of scattering simulations and P-RSoXS reveals information on the chain order parameter, the length that the chain is oriented, and the rate of decay of the orientational order from the nanoparticle surface. Obtaining this level of detail on chain shape near nanoparticle surfaces is unprecedented and represents a remarkable advance in the ability to characterize amorphous chains near surfaces. Furthermore, this paper resolves unresolved questions in the field of X-ray scattering on the origins of scattering patterns that result from interactions between polarized soft X-rays and local molecular orientation correlations. I suggest a few edits prior to publication.

One of the achievements of this work is in demonstrating the sensitivity to different grafting densities, for example when comparing AuPS27 and AuPS53. However, it is important for the authors to show whether the differences in the reported parameters for these two samples are statistically significant, based on the error of their fits to the simulations. The authors have performed a very nice uncertainty analysis that should allow them to report on whether the differences in S_0 , raniso , and d between AuPS27 and AuPS53 are significant based on sensitivity of the overall fit to variations of each of the parameters. It would also be nice to report an uncertainty in the individual parameters, if possible. Note that if these are reported as errors, they should be clearly denoted as uncertainties of fits, and not as standard error from multiple measurements.

It appears that the authors are demonstrating how the anisotropy arises from correlations between local chain orientations, or what we often term the structure factor component of scattering data. This resolves an open question in the field, of whether anisotropy arises from the shape of nanostructure, or what is often termed the form factor, or whether anisotropy arises from correlations between locally-oriented domains. Perhaps more accurately, the authors can address whether scattering anisotropy in RSoXS arises from correlations in orientation within a domain (structure factor between chains) or from correlations in orientation between domains (structure factor between graft/nanoparticles), or from both. Can the authors comment on whether the contribution to the anisotropy from each individual particle is significant to the overall scattering anisotropy, especially when normalized by the scattering volume?

The authors clearly show that their results are not consistent with the predicted scattering from tangential chains, or chains parallel to the nanoparticle surface. The authors state that “tangential chains, which are sometimes reported, are not present here” (page 14). This is perhaps too strong of a statement. Is it possible that the order parameter being less than 1 is a consequence of mixed orientations, where some chains are tangential and some are extended? Or can the authors completely eliminate the possibility of any tangential chains?

Reviewer #2 (Remarks to the Author):

In the manuscript, authors report on a calculation method combining GPU acceleration to decipher

the molecular orientation of grafted polymers. The theoretical understanding of the Polarized RSoXS is accurate. Taking advantage of the orientation recognition ability of RSoXS, authors raise a straightforward way to interpret the scattering data in nanoscale. The calculation approach is convincing. Authors also raise some potential problems regarding to the computation. Considering the importance of learning the orientation distribution of confined polymers, this manuscript could have interests to broad communities. However, I do have some concerns on how universal could this method be applied to and how to further explain the fitting parameters (anisotropy, S_0 and d) properly.

1. Authors applied an image analysis algorithm to the AFM images and established a model for further computation. I would like to ask, did AFM and Polarized RSoXS used the same sample? To my understanding, the computation is actually a FT of the model with extraordinary axis field defined by $S(r)$. If so, the position of gold NPs is also an impact factor, which requires identical between image analysis and computation.

2. In the modelling part, which is the most critical one for computation, authors used NEXAFS to determine the highlighted functional group. This is quite convincing to me and I think such procedure can be extended to other absorption edges. What if there are multiple components orientated, which is quite a common problem in resonant scattering study for small molecules. Is there any better way for modelling?

3. Concerning the further application of such a precise computation, is there any possibility to prove the obtained orientation by other methods?

4. I think there is more to do to really understand the fitting parameters and their dependence to systems comprehensively. For example, a simple question might be raised, did the size distribution affects fitting? And how severe the influence might be?

In conclusion, though a few questions need to be solved, the computation is well done and could be of high impact. Therefore, I would recommend this manuscript to be published in Nature Communications if all my concerns could be answered properly.

Reviewer #3 (Remarks to the Author):

This manuscript utilized resonant soft x-ray scattering to study polymer-grafted nanoparticle composites. This work's key novelty lies in that it takes into consideration the orientation dependence in the scattering process, an interplay between polarized x-rays and energy-dependent anisotropy of refractive indices. Consequently, the work demonstrates that it is possible to access unique information related to orientation ordering near the nanoparticle surfaces and how it decays off the nanoparticle surface into the corona. Such info is highly desirable but has been challenging to obtain using other characterization tools. Using prior information from AFM and absorption spectroscopy measurements, the scattering pattern simulation employed a GPU-based approach to cover an ample parameter space, which turns out to be a potent and effective tool in identifying a reasonable solution, a significant step forward. This approach may broadly benefit the areas of polymer, soft matter, and even biomaterials, where bond orientation plays a critical role.

Thus, I would recommend acceptance of this manuscript after the authors address the following questions:

1. motivation and scientific impact. On a high level, the ability to access bond orientation order parameters is undoubtedly crucial. However, it would be good to provide more specifics when introducing the motivation and potential impact of this work, i.e. real-world applications or other fundamental scientific problems may directly benefit from this information. The current introduction

is a bit too brief on the science side.

2. Line 94. "2D hexagonal close packed lattice at $q \approx 0.15 \text{ nm}^{-1}$ (average interparticle spacing $\approx 40 \text{ nm}$ ". Plz define 'interparticle spacing', is it the average center-to-center distance between nearest neighbors? Or is it the layer spacing of the fundamental SAXS peak? It seems these two are used as interchangeable in this manuscript. For a typical 2-d hexagonal lattice, the layer spacing of the first peak Q_{10} or Q_{01} corresponds to a d-spacing that equals to $\sqrt{3}/2 * a$, where a is the center-to-center distance between nearest neighbors. Please clarify.

3. Fig. 1b,c, and caption, suggest using the same unit for Q throughout this manuscript, i.e. either nanometer or angstrom, instead of both in different places.

4. Line 103: "We confirmed that $A(q)$ was identical for two orthogonal directions of the electric field of the incident X-rays (Figure S3), demonstrating the absence of any nonisotropic/non-uniaxial assembly in the films". Fig. S3 only shows the Intensity vs Q for horizontal direction in (a) and vertical direction in (b), without showing any $A(q)$. so it is not clear whether $A(q)$ is identical as claimed? The two curves in fig. S3c are quantitatively not identical; therefore, it is unclear whether the slight difference is significant or not in $A(q)$ calculation.

5. Fig. 5b and Fig. S6. Compared to the experimental data, the first peak in simulation is at a very similar q while the second peak is at a much higher q , which is a bit strange. Please comment.

6. The samples studied here include nanoparticles coated with different lengths of polymers, grafted at different densities. It seems it would be good to have at least one sample, which has the same type of polymer but different grafting density, i.e., varying either polymer length or grafting density, one at a time.

RESPONSE TO REVIEWERS

Individual reviewer comments are addressed below.

Reviewer #1 (Remarks to the Author):

This manuscript reports on developing an approach based on polarized resonant soft X-ray scattering (P-RSoXS) to examine the local alignment of chains grafted to nanoparticle surfaces. The combination of scattering simulations and P-RSoXS reveals information on the chain order parameter, the length that the chain is oriented, and the rate of decay of the orientational order from the nanoparticle surface. Obtaining this level of detail on chain shape near nanoparticle surfaces is unprecedented and represents a remarkable advance in the ability to characterize amorphous chains near surfaces. Furthermore, this paper resolves unresolved questions in the field of X-ray scattering on the origins of scattering patterns that result from interactions between polarized soft X-rays and local molecular orientation correlations. I suggest a few edits prior to publication.

We thank the reviewer for this positive appraisal of the merits of the work.

One of the achievements of this work is in demonstrating the sensitivity to different grafting densities, for example when comparing AuPS27 and AuPS53. However, it is important for the authors to show whether the differences in the reported parameters for these two samples are statistically significant, based on the error of their fits to the simulations. The authors have performed a very nice uncertainty analysis that should allow them to report on whether the differences in S_0 , raniso , and d between AuPS27 and AuPS53 are significant based on sensitivity of the overall fit to variations of each of the parameters. It would also be nice to report an uncertainty in the individual parameters, if possible. Note that if these are reported as errors, they should be clearly denoted as uncertainties of fits, and not as standard error from multiple measurements.

We thank the reviewer for pointing out this omission. As a National Measurement Institute, uncertainties are very important to us. We have added uncertainties in the individual parameters from the variance-covariance matrix obtained from the heat map data in Figure 4. Details of the calculation approach are provided in Methods.

It appears that the authors are demonstrating how the anisotropy arises from correlations between local chain orientations, or what we often term the structure factor component of scattering data. This resolves an open question in the field, of whether anisotropy arises from the shape of nanostructure, or what is often termed the form factor, or whether anisotropy arises from correlations between locally-oriented domains. Perhaps more accurately, the authors can address whether scattering anisotropy in RSoXS arises from correlations in orientation within a domain (structure factor between chains) or from correlations in orientation between domains (structure factor between graft/nanoparticles), or from both. Can the authors comment on whether the contribution to the anisotropy from each individual particle is significant to the overall scattering anisotropy, especially when normalized by the scattering volume?

We enthusiastically thank the reviewer for this comment, which has led to a new discussion in the manuscript (p16-17) and new results that may be of additional value to the community. Simulations with different numbers of particles were done, from the dilute limit to the well-packed morphologies. The

results (in SI) show clearly that the anisotropy arises from “correlations in orientation within a domain”, as the reviewer puts it, and it is largely independent of number of particles. Although we had a rough sense that this was true, we were surprised by how little the structure factor influences the result. We conclude our discussion in the manuscript as follows:

These simulation results show that $A(q)$ vs. q has special value as an objective function for fitting P-RSoXS data because in densely packed systems it is largely insensitive to the structure factor. In model morphologies where $I(q)$ vs. q does not completely resemble the experimental data (because, perhaps, a single AFM image is not representative of the ensemble measured by scattering), fitting $A(q)$ vs. q will still extract orientation parameters relevant to orientation in the corona of individual particles.

The authors clearly show that their results are not consistent with the predicted scattering from tangential chains, or chains parallel to the nanoparticle surface. The authors state that “tangential chains, which are sometimes reported, are not present here” (page 14). This is perhaps too strong of a statement. Is it possible that the order parameter being less than 1 is a consequence of mixed orientations, where some chains are tangential and some are extended? Or can the authors completely eliminate the possibility of any tangential chains?

We thank the referee for pointing this out. The measurement cannot exclude some fraction of tangential chains. But it is also not just a simple average. We have substantially revised the text on p15 to explain the situation.

Reviewer #2 (Remarks to the Author):

In the manuscript, authors report on a calculation method combining GPU acceleration to decipher the molecular orientation of grafted polymers. The theoretical understanding of the Polarized RSoXS is accurate. Taking advantage of the orientation recognition ability of RSoXS, authors raise a straightforward way to interpret the scattering data in nanoscale. The calculation approach is convincing. Authors also raise some potential problems regarding to the computation. Considering the importance of learning the orientation distribution of confined polymers, this manuscript could have interests to broad communities.

We thank the reviewer for suggesting the importance of this work to broad communities.

However, I do have some concerns on how universal could this method be applied to and how to further explain the fitting parameters (anisotropy, S_0 and d) properly.

1. Authors applied an image analysis algorithm to the AFM images and established a model for further computation. I would like to ask, did AFM and Polarized RSoXS used the same sample? To my understanding, the computation is actually a FT of the model with extraordinary axis field defined by $S(r)$. If so, the position of gold NPs is also an impact factor, which requires identical between image analysis and computation.

Yes, the same sample was used for the AFM. We have added this information in the experimental section. In response to this question and reviewer #1's question about form factor vs. structure factor effects, we have also added additional discussion p17 about how a single AFM image may not be representative of

the ensemble measured by scattering, but fitting $A(q)$ vs. q will still extract orientation parameters relevant to orientation in the corona of individual particles.

2. In the modelling part, which is the most critical one for computation, authors used NEXAFS to determine the highlighted functional group. This is quite convincing to me and I think such procedure can be extended to other absorption edges. What if there are multiple components orientated, which is quite a common problem in resonant scattering study for small molecules. Is there any better way for modelling?

We thank the reviewer for their prescient question. This is indeed the direction we intend to develop the technique! Building on the approach we communicate here, experiments with multiple components will need to take advantage of the spectroscopic chemical sensitivity of the technique across a wider energy range and across multiple absorption edges. Our approach can be extended to these problems; the current version of our forward-simulation software allows for up to five materials that have unique chemistries and can simulate patterns across any edge, even into tender X-ray ranges. We have added comments in the introduction to emphasize these possible growth directions.

3. Concerning the further application of such a precise computation, is there any possibility to prove the obtained orientation by other methods?

This question really goes to the heart of why we are pursuing this technique – there's not really another method that measures the spatial distribution of molecular orientation at nanoscopic length scales. Modern TEM techniques such as 4D-STEM can be used on semicrystalline materials, but in systems such as the grafted nanoparticles in the manuscript under review there is no order that would give rise to local diffraction and 4D-STEM cannot measure the orientation. Molecular modeling, either coarse-grained or atomistic, is likely to produce results most relevant to our measurement. We anticipate that RSoXS will become a means to validate such molecular models and we are working with leading groups in this area to make the connections.

4. I think there is more to do to really understand the fitting parameters and their dependence to systems comprehensively. For example, a simple question might be raised, did the size distribution affects fitting? And how severe the influence might be?

Besides the simpler 3-parameter model we carried out simulations on a 5-parameter model where we additionally varied core size variation (from SEM) and surface roughness (from AFM). We have discussed these studies alongside a detailed uncertainty analysis in section 15 of Supporting Information. Although we obtained a more “realistic” morphology by including the two additional parameters, we observed a marginal decrease in SSE (1.3 to 1.2) without any noticeable changes in the fits. The error distribution for variation of all 5 parameters is shown in Figure S16 in the SI. The one-sided nature of the error distribution shows that the optimal values obtained from the 3-parameter model are still valid for the 5-parameter model. However, we believe that with development of better detectors with lower noise floor might enable the study of such subtler effects. We have now added some text and refer to these figures more explicitly in the manuscript.

In conclusion, though a few questions need to be solved, the computation is well done and could be of high impact. Therefore, I would recommend this manuscript to be published in Nature Communications if

all my concerns could be answered properly.

Reviewer #3 (Remarks to the Author):

This manuscript utilized resonant soft x-ray scattering to study polymer-grafted nanoparticle composites. This work's key novelty lies in that it takes into consideration the orientation dependence in the scattering process, an interplay between polarized x-rays and energy-dependent anisotropy of refractive indices. Consequently, the work demonstrates that it is possible to access unique information related to orientation ordering near the nanoparticle surfaces and how it decays off the nanoparticle surface into the corona. Such info is highly desirable but has been challenging to obtain using other characterization tools. Using prior information from AFM and absorption spectroscopy measurements, the scattering pattern simulation employed a GPU-based approach to cover an ample parameter space, which turns out to be a potent and effective tool in identifying a reasonable solution, a significant step forward. This approach may broadly benefit the areas of polymer, soft matter, and even biomaterials, where bond orientation plays a critical role.

Thus, I would recommend acceptance of this manuscript after the authors address the following questions:

1. motivation and scientific impact. On a high level, the ability to access bond orientation order parameters is undoubtedly crucial. However, it would be good to provide more specifics when introducing the motivation and potential impact of this work, i.e. real-world applications or other fundamental scientific problems may directly benefit from this information. The current introduction is a bit too brief on the science side.

We thank the reviewer for this suggestion! We have added a new paragraph in the Introduction section that addresses how this method can be useful to a broad range of systems.

2. Line 94. “2D hexagonal close packed lattice at $q \approx 0.15 \text{ nm}^{-1}$ (average interparticle spacing $\approx 40 \text{ nm}$)”. Plz define ‘interparticle spacing’, is it the average center-to-center distance between nearest neighbors? Or is it the layer spacing of the fundamental SAXS peak? It seems these two are used as interchangeable in this manuscript. For a typical 2-d hexagonal lattice, the layer spacing of the first peak Q_{10} or Q_{01} corresponds to a d-spacing that equals to $\sqrt{3}/2 * a$, where a is the center-to-center distance between nearest neighbors. Please clarify.

We thank the reviewer for catching this inconsistency. We have revised accordingly to avoid any ambiguity.

The numbers correspond to median d-spacing corresponding to fundamental SAXS peak. Since it is a paracrystalline 2-d hcp lattice, not perfect, the above relation between the lattice parameter and d-spacing is not strictly applicable here. However, the match between the SAXS peak positions in experiment and simulation shows that the AFM images capture interparticle spacings that are representative of the volume measured by RSoXS. We now stick to interparticle spacings from AFM when discussing real space.

3. Fig. 1b,c, and caption, suggest using the same unit for Q throughout this manuscript, i.e. either nanometer or angstrom, instead of both in different places.

We thank the reviewer for catching this inconsistency. q-scale in figures have been changed to nm^{-1} .

4. Line 103: “We confirmed that $A(q)$ was identical for two orthogonal directions of the electric field of the incident X-rays (Figure S3), demonstrating the absence of any nonisotropic/non-uniaxial assembly in the films”. Fig. S3 only shows the Intensity vs Q for horizontal direction in (a) and vertical direction in (b), without showing any $A(q)$. so it is not clear whether $A(q)$ is identical as claimed? The two curves in fig. S3c are quantitatively not identical; therefore, it is unclear whether the slight difference is significant or not in $A(q)$ calculation.

$A(q)$ is difference over sum of intensities in the parallel and perpendicular directions and is independent of the scattered intensity. Nevertheless, we have included the $A(q)$ profiles corresponding to horizontal and vertical polarization in Figure S3 in the SI. Since they are identical all analysis manuscript was done using the average of the two profiles at all measured energies for both the samples. We have now made this clear in the manuscript.

5. Fig. 5b and Fig. S6. Compared to the experimental data, the first peak in simulation is at a very similar q while the second peak is at a much higher q , which is a bit strange. Please comment.

In response to this question and reviewer #1’s question about form factor vs. structure factor effects, we have added additional discussion p17 about how a single AFM image may not be representative of the ensemble measured by scattering, but fitting $A(q)$ vs. q will still extract orientation parameters relevant to orientation in the corona of individual particles.

6. The samples studied here include nanoparticles coated with different lengths of polymers, grafted at different densities. It seems it would be good to have at least one sample, which has the same type of polymer but different grafting density, i.e., varying either polymer length or grafting density, one at a time.

Unfortunately, the grafting density and molecular weight are closely correlated due to the “grafting-to” approach we use to make polymer-grafted nanoparticles We regret that it is not possible to create the proposed particles with the current authoring team.

REVIEWERS' COMMENTS

Reviewer #1 (Remarks to the Author):

The authors have very nicely addressed all concerns.

Reviewer #2 (Remarks to the Author):

I am satisfied with the modifications made.

Reviewer #3 (Remarks to the Author):

These authors have reasonably addressed the reviewers' comments, therefore, I would recommend its publication in Nat. Comm.